# Precedent-Enhanced Legal Judgment Prediction with LLM and Domain-Model Collaboration

**Yiquan Wu[1], Siying Zhou[1], Yifei Liu[1], Weiming Lu[1*], Xiaozhong Liu[2]**
**Yating Zhang[3], Changlong Sun[13], Fei Wu[1*], Kun Kuang[1*]**

[1]Zhejiang University, Hangzhou, China
[2]Worcester Polytechnic Institute, USA
[3]Alibaba Group, Hangzhou, China

{wuyiquan, zhousiying, liuyifei, luwm, kunkuang}@zju.edu.cn, yatingz89@gmail.com

xliu14@wpi.edu, changlong.scl@taobao.com, wufei@cs.zju.edu.cn

## Abstract

Legal Judgment Prediction (LJP) has become an increasingly crucial task in Legal AI, i.e., predicting the judgment of the case in terms of case fact description. Precedents are the previous legal cases with similar facts, which are the basis for the judgment of the subsequent case in national legal systems. Thus, it is worthwhile to explore the utilization of precedents in the LJP. Recent advances in deep learning have enabled a variety of techniques to be used to solve the LJP task. These can be broken down into two categories: large language models (LLMs) and domain-specific models. LLMs are capable of interpreting and generating complex natural language, while domain models are efficient in learning task-specific information. In this paper, we propose the precedent-enhanced LJP framework (PLJP) – a system that leverages the strength of both LLM and domain models in the context of precedents. Specifically, the domain models are designed to provide candidate labels and find the proper precedents efficiently, and the large models will make the final prediction with an in-context precedents comprehension. Experiments on the real-world dataset demonstrate the effectiveness of our PLJP. Moreover, our work shows a promising direction for LLM and domain-model collaboration that can be generalized to other vertical domains.

## 1 Introduction

Legal AI has been the subject of research for several decades, with the aim of assisting individuals in various legal tasks, including legal QA (Monroy et al., 2009), court view generation (Wu et al., 2020), legal entity recognition (Cardellino et al., 2017), and so on. As one of the most important legal tasks, legal judgment prediction (LJP) aims to predict the legal judgment of the case based on the case fact description. The legal judgment typically includes the law article, charge and prison term.

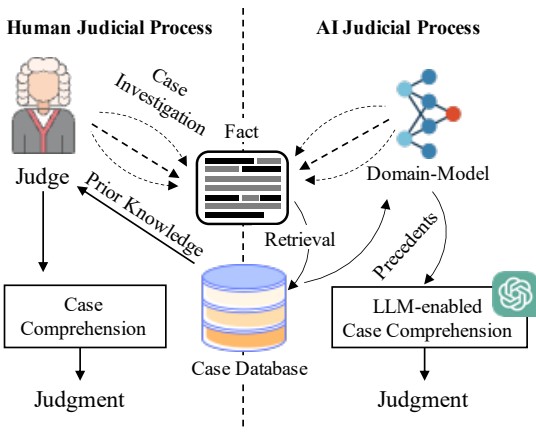

Figure 1: An illustration of the judicial process, our motivation is to promote the collaboration between the domain model and LLM (right part) for simulating the judicial process of the human judge (left).

Precedents, which refer to previous cases with similar fact descriptions, hold a crucial position within national legal systems (Guillaume, 2011). On a more macro level, precedents are known as the collective body of judge-made laws in a nation(Garner, 2001). They serve the purpose of ensuring consistency in judicial decisions, providing greater legal guidance to judges and facilitating legal progress and evolution to meet dynamic legal demands. In the Common Law system, the precedents are the mandatory basis of the judgment of the subsequent case (Rigoni, 2014). In the Civil Law system, judge-made laws are perceived as secondary legal sources while written laws are the basic legal sources(Larenz, 1992). In the contemporary era, there is also a growing trend to treat the precedents as a source of "soft" law (Fon and Parisi, 2006), and judges are expected to take them into account when reaching a decision (Guillaume, 2011). Thus, it is worthwhile to explore the utilization of precedents in the legal judgment prediction.

With the development of deep learning, many technologies have been adopted in the LJP task, which can be split into two categories: large lan-

---

[*]Corresponding Authors.

guage models (LLMs) and domain-specific models (Ge et al., 2023). Owing to extensive training, LLMs are good at understanding and generating complex natural language, as well as in-context learning. On the other hand, domain-specific models are designed to cater to specific tasks and offer cost-effective solutions. However, when it comes to incorporating precedents into the LJP task, both categories of models face certain limitations. LLMs, constrained by their prompt length, struggle to grasp the meaning of numerous abstract labels and accurately select the appropriate one. For domain models, though trained with label annotations, the drawback is the limited ability to comprehend and distinguish the similarities and differences between the precedents and the given case.

In this paper, as Fig. 1 shows, we try to collaborate the LLMs with the domain-specific models and propose a novel precedent-enhanced legal judgment prediction framework (PLJP). Specifically, domain models contribute by providing candidate labels and finding the proper precedents from the case database effectively; the LLMs will decide the final prediction through an in-context precedent comprehension.

Following the previous LJP works (Zhong et al., 2018; Yue et al., 2021; Dong and Niu, 2021), our experiments are conducted on the publicly available real-world legal dataset. To prevent any potential data leakage during the training of the LLMs, where the model may have already encountered the test cases, we create a new test set comprising cases that occurred after 2022. This is necessary because the LLMs we utilize have been trained on a corpus collected only until September 2021. By doing so, we ensure a fair evaluation of the PLJP framework. Remarkably, our proposed PLJP framework achieves state-of-the-art (SOTA) performance on both the original test set and the additional test set.

To sum up, our main contributions are as follows:

- We address the important task of legal judgment prediction (LJP) by taking precedents into consideration.

- We propose a novel precedent-enhanced legal judgment prediction (PLJP) framework that leverages the strength of both LLM and domain models.

- We conduct extensive experiments on the real-world dataset and create an additional test set

to ensure the absence of data leakage during LLM training. The results obtained on both the original and additional test sets validate the effectiveness of the PLJP framework.

- Our work shows a promising direction for LLM and domain-model collaboration that can be generalized over vertical domains. We make all the codes and data publicly available to motivate other scholars to investigate this novel and interesting research direction[1].

## 2 Related Work

### 2.1 Legal AI

Legal Artificial Intelligence (Legal AI) aims to enhance tasks within the legal domain through the utilization of artificial intelligence techniques (Zhong et al., 2020; Katz et al., 2023). Collaborative efforts between researchers in both law and computer fields have been lasting to explore the potential of Legal AI and its applications across various legal tasks. These tasks encompass areas such as legal question answering (QA) (Monroy et al., 2009), legal entity recognition (Cardellino et al., 2017), court view generation (Wu et al., 2020), legal summarization (Hachey and Grover, 2006; Bhattacharya et al., 2019), legal language understanding(Chalkidis et al., 2022) and so on.

In this work, we focus on the task of legal judgment prediction, which is one of the most common tasks in Legal AI.

### 2.2 Legal Judgment Prediction

Legal judgment prediction (LJP) aims to predict judgment results based on the fact descriptions automatically (Lin et al., 2012; Chalkidis et al., 2019; Yue et al., 2021; Xu et al., 2020; Niklaus et al., 2021; Malik et al., 2021; Feng et al., 2022; Lyu et al., 2022; Gan et al., 2022). The LJP methods in earlier years required manually extracted features (Keown, 1980), which is simple but costly. Owing to the prosperity of machine learning (Wu et al., 2022; Shen et al., 2022; Li et al., 2022a,b; Zhang et al., 2022; Li et al., 2023; Zhang et al., 2023), researchers began to formalize the LJP problem with machine learning methods. These data-driven methods can learn the features with far less labor (e.g., only the final labels are required). Sulea et al. (2017) developed an ensemble system that averages the output of multiple SVM to improve the

[1]https://github.com/wuyiquan/PLJP

performance of LJP. Luo et al. (2017) utilized an attention mechanism in the LJP. Zhong et al. (2018) considered the dependency of the sub-tasks in the LJP. Yue et al. (2021) investigated the problem by separating the representation of fact description into different embedding. Liu et al. (2022) used contrastive learning in the LJP.

However, these existing LJP methods tend to overlook the significance of precedents. In this study, we propose a precedent-enhanced LJP framework (PLJP) that leverages the collaboration between domain-specific models and large language models (LLMs) to address the LJP task.

## 2.3 Precedent Retrieval

The precedent is the basis of judgment in the Common Law system, and also an important reference for decision-making in the Civil Law system. Therefore, precedent retrieval is another valuable task in Legal AI (Althammer et al., 2021). There are two main precedent retrieval models: expert knowledge-based models and natural language processing (NLP)-based models (Bench-Capon et al., 2012). Expert knowledge-based models use the designed sub-elements to represent the legal cases (Saravanan et al., 2009), while NLP-based models mainly convert the text into embeddings and then calculate the similarity from the embedding level (Ma et al., 2021; Chalkidis et al., 2020).

Most retrieval models required additional annotation so can not be directly applied to the LJP task. In our paper, we use an unsupervised dense retrieval model (Izacard et al., 2022) to get the precedents, which can be updated by other retrieval models if needed.

## 2.4 Large Language Models

Large language models (LLMs), such as ChatGPT, have attracted widespread attention from society (Zhao et al., 2023). With pre-training over large-scale corpora, LLMs show strong capabilities in interpreting and generating complex natural language, as well as reasoning (e.g., in-context learning). The technical evolution of LLMs has been making an important impact on the fields of natural language processing (Brown et al., 2020; Touvron et al., 2023), computer vision (Shao et al., 2023; Wu et al., 2023), and reinforcement learning (Du et al., 2023). In the legal domain, LLMs can also be used for many tasks such as legal document analysis and legal document writing (Sun, 2023).

However, in the prediction tasks, which can involve dozens of abstract labels, the performance of LLMs is not as good as in generation tasks, due to the limited prompt length. In this paper, we explore the utilization of LLMs in the LJP task with the collaboration of domain-specific models.

## 3 Problem Formulation

In this work, we focus on the problem of legal judgment prediction. We first clarify the definition of the terms as follows.

• **Fact Description** refers to a concise narrative of the case, which typically includes the timeline of events, the actions or conduct of each party, and any other essential details that are relevant to the case. Here we define it as a token sequence $f = \{w_t^f\}_{t=1}^{l_f}$, where $l_f$ is the length.

• **Judgment** is the final decision made by a judge in a legal case based on the facts and the precedents. It typically consists of the law article, the charge, and the prison term. We represent the judgment of a case as $j = (a, c, t)$, where $a, c, t$ refer to the labels of article, charge and prison term, respectively.

• **Precedent** is the previous case with a similar fact. The judgments of the precedents are important references for the current case. Here, a precedent is defined as $p = (f_p, j_p)$, where $f_p$ is its fact description and $j_p$ is its judgment. For a given case, there can be several precedents, which can be denoted as $P = \{p_1, p_2, ..., p_n\}$, where $n$ is the number of precedents.

Then the problem can be defined as:

**Problem 1** (Legal Judgment Prediction). *Given the fact description $f$, our task is to get and comprehend the precedents $P$, then predict the judgment $j = (a, c, t)$.*

## 4 Precedent-Enhanced LJP (PLJP)

In this section, we describe our precedent-enhanced legal judgment prediction framework (PLJP), Fig. 2 shows the overall framework.

## 4.1 Case Database Construction

Before we use the precedents, we have to collect a large number of previous cases to construct a case database. Since the fact descriptions are usually long and elaborate, it is difficult for the models to get the proper precedents. To this end, we reorganize the fact description of these previous cases with the help of LLMs.

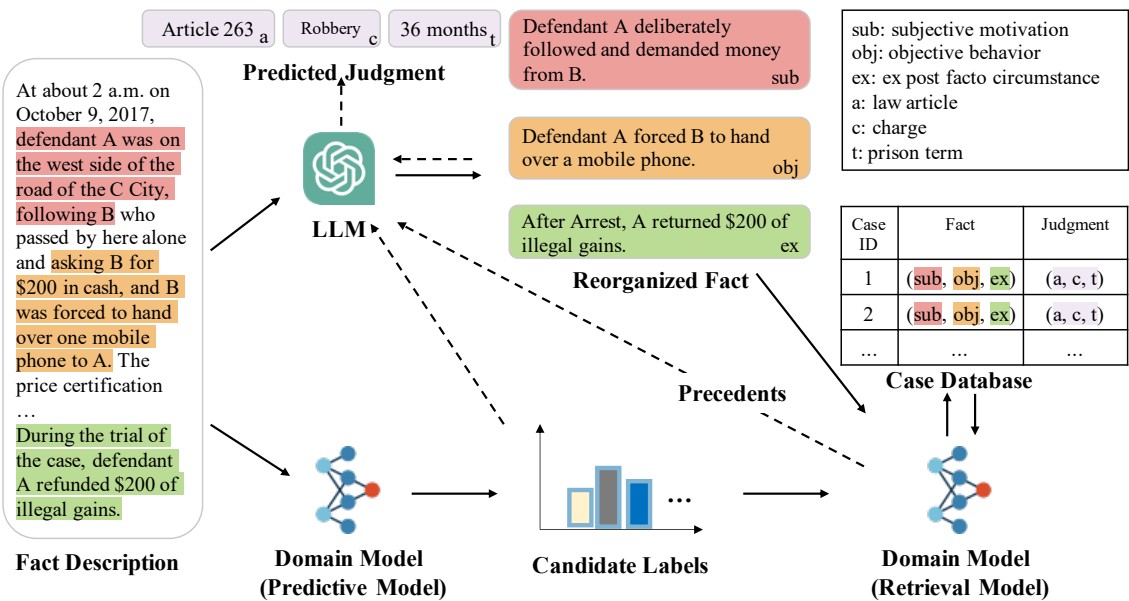

Figure 2: The overall framework of PLJP, where the sub, obj and ex refer to the subjective motivation, objective behavior and ex post facto circumstance, respectively. The solid lines are the precedent retrieval process, while the dotted lines represent the process of the prediction.

### 4.1.1 Fact Reorganization

Given a fact description of a case, we summarize it from three aspects: subjective motivation, objective behavior, and ex post facto circumstances. The reorganization doesn't require human annotation and is completed by the LLMs with the following prompts: *"A fact description can be categorized into subjective motivation, objective behavior, and ex post facto circumstances. Subjective motivation refers to the psychological attitude of the perpetrator towards their harmful actions and their consequences, including intent, negligence, and purposes of the crime. Objective behavior pertains to the necessary conditions for constituting a crime in terms of observable activities, including harmful conduct, harmful results, and the causal relationship between the conduct and the results. Ex post facto circumstances are various factual situations considered when determining the severity of penalties. Mitigating circumstances for lenient punishment include voluntary surrender and meritorious conduct, while aggravating circumstances for harsher punishment include recidivism. Based on the provided information, your task is to summarize the following facts."*

The reorganization reduces the length of facts and makes the precedents easy to get and comprehend in the PLJP.

After the reorganization, the fact description $f$ is translated to a triplet $(sub, obj, ex)$, which indi-

cates the subjective motivation, objective behavior, and ex post facto circumstances, respectively. Finally, a previous case in the case database is stored as a pair of reorganized facts and the judgment.

### 4.2 Legal Judgment Prediction

Next, we describe the collaboration of the LLM and domain models in legal judgment prediction.

### 4.2.1 Domain Models

The domain models are trained on specific datasets, aiming to solve certain tasks. Here, we use two kinds of domain models, including the predictive model and the retrieval model.

**Predictive model.** The predictive model takes the fact description as the input and outputs the candidate labels of the three sub-tasks (e.g., law article, charge, prison term). Since the fact description $f = \{w_t^f\}_{t=1}^{l_f}$ are sequences of words, we first transform it into embedding sequence $H^f \in \mathbb{R}^{l_f \times d}$ with an Encoder:

$$H^f = \text{Encode}(f), \qquad (1)$$

where $H^f = h_1^f, h_2^f, ..., h_{l_f}^f$, and $d$ is the dimension of the embedding.

We take a max-pooling operation to obtain the pooled hidden vector $h^f \in \mathbb{R}^d$ and then feed it into a fully-connected network with softmax activation

to obtain the label probability distribution $P \in \mathbb{R}^m$:

$$h^f = \text{MaxPooling}(H^f),$$
$$P = \text{Softmax}(W^p \cdot h^f + b^p), \qquad (2)$$

where $W^p \in \mathbb{R}^{m \times d}$ and $b^p \in \mathbb{R}^m$ are learnable parameters. Note $m$ varies in different sub-tasks.

Then, each sub-task gets its candidate labels according to the probability distribution $P$, and the number of candidate labels is equal to the number of precedents $n$.

**Retrieval model.** The retrieval model aims to get the proper precedents of the given case based on its reorganized fact $(sub, obj, ex)$.

Formally, to get the similarity score of any two texts $D_1$ and $D_2$, we will first encode each of them independently using the same encoder:

$$h_{D_1} = \text{Encoder}(D_1), h_{D_2} = \text{Encoder}(D_2), \qquad (3)$$

where $h_{D_1} \in \mathbb{R}^{d'}$ and $h_{D_2} \in \mathbb{R}^{d'}$ are the embedding of each, $d'$ is the dimension. The similarity score $s(D_1, D_2)$ is then the cosine similarity of the $h_{D_1}$ and $h_{D_2}$:

$$s(D_1, D_2) = \frac{h_{D_1} \cdot h_{D_2}}{\|h_{D_1}\| \|h_{D_2}\|}. \qquad (4)$$

Here we concatenate the $sub$, $obj$ and $ex$ into a whole text to calculate the similarity score of the given case and the cases in the case database.

For each candidate label, we pick one case as the precedent: the case that has the highest similarity score and has the same label. For example, if the label "Theft" is in the candidate labels in the charge prediction, we will find the most similar previous case with the same label as the corresponding precedent. The one-to-one relationship between the candidate label and precedent helps the LLM distinguish the differences among the labels. In other words, the precedent serves as a supplementary explanation of the label.

Finally, we get precedents $P = \{p_1, p_2, ..., p_n\}$ for the given case.

### 4.2.2 LLMs

The large language models are models with billions of parameters, which are trained on large-scale corpora, and show strong capabilities in interpreting and generating complex natural language. LLMs contribute to PLJP by fact reorganization and in-context precedent comprehension.

**Fact Reorganization** The fact reorganization is described in case database construction (Sec. 4.1.1), which aims to summarize the fact description from three aspects by the LLMs. Besides the database contribution, as Fig. 2 shows, when a new test case comes, the LLMs will reorganize the fact description with the same prompt.

**In-Context Precedent Comprehension** Since LLMs are capable of understanding complex natural language, we stack the given case with its precedents and let the LLMs make the final prediction by an in-context precedent comprehension. Specifically, the prompt of law article prediction is designed as follows: *"Based on the facts, we select the candidate law articles by the domain models and select the following three precedents based on the candidate law articles. Please comprehend the difference among the precedents, then compare them with the facts of this case, and choose the final label."*

Consider the topological dependencies among the three sub-tasks (Zhong et al., 2018), in the prediction of charge, we add the predicted law article in the prompt; in the prediction of prison term, we add the predicted law article and charge.

### 4.3 Training

In PLJP, considering the realizability, we train domain models on legal datasets and leave the LLMs unchanged. To train predictive models, the cross-entropy loss is employed. As for retrieval models, contrastive loss is used like Izacard et al. (2022).

## 5 Experiments

| Type | CAIL2018 | CJO22 |
|------|----------|-------|
| # Law Article | 164 | 164 |
| # Charge | 42 | 42 |
| # Prison Term | 10 | 10 |
| # Sample | 82138 | 1698 |
| Avg. # words in Fact | 288.6 | 461.7 |

Table 1: Statistics of datasets.

### 5.1 Datasets

Following many influential LJP works (Zhong et al., 2018; Xu et al., 2020; Yue et al., 2021; Dong and Niu, 2021), our experiment is conducted on the widely used and publicly available CAIL2018 dataset, which is a Chinese dataset in the context of People's Republic of China (PRC). This dataset consists of real-world cases, each of which

| Method | CJO22 | | | | CAIL2018 | | | |
|---|---|---|---|---|---|---|---|---|
| | Acc | Ma-P | Ma-R | Ma-F | Acc | Ma-P | Ma-R | Ma-F |
| CNN (LeCun et al., 1989) | 76.14 | 35.48 | 38.55 | 35.39 | 80.50 | 40.10 | 38.33 | 38.49 |
| BERT (Devlin et al., 2019) | 82.62 | 45.89 | 47.91 | 45.83 | 82.77 | 36.82 | 35.94 | 35.82 |
| Roberta (Liu et al., 2019) | 80.32 | 42.36 | 44.22 | 41.80 | 83.08 | 48.09 | 44.25 | 44.87 |
| TopJudge (Zhong et al., 2018) | 78.73 | 40.38 | 41.47 | 40.09 | 80.46 | 40.96 | 40.96 | 38.24 |
| R-Former (Dong and Niu, 2021) | 87.69 | 53.03 | 49.35 | 50.23 | **87.82** | 56.13 | 56.57 | 55.81 |
| LADAN (Xu et al., 2020) | 79.44 | 48.43 | 44.13 | 46.18 | 82.82 | 42.57 | 39.00 | 40.71 |
| NeurJudge (Yue et al., 2021) | 71.38 | 52.86 | 53.52 | 52.62 | 76.91 | 55.95 | 52.92 | 53.56 |
| EPM(Feng et al., 2022) | 84.19 | 47.21 | 43.79 | 44.39 | 85.80 | 49.08 | 45.76 | 47.32 |
| CTM(Liu et al., 2022) | 79.44 | 47.83 | 42.25 | 43.43 | 84.72 | 46.46 | 44.83 | 45.10 |
| Dav003 | 2.10 | 0.82 | 0.17 | 0.26 | 1.02 | 0.30 | 0.08 | 0.13 |
| 3.5turbo | 9.13 | 2.54 | 1.61 | 1.53 | 4.08 | 4.95 | 3.64 | 2.30 |
| PLJP(CNN) | 87.67 | 55.21 | 55.59 | 54.37 | 86.05 | 58.08 | 56.46 | 54.92 |
| PLJP(BERT) | **94.18** | **74.65** | **76.23** | **74.84** | 87.07 | **58.81** | **57.29** | **56.63** |

Table 2: Results of law article prediction, the best is **bolded** and the second best is underlined.

| Method | CJO22 | | | | CAIL2018 | | | |
|---|---|---|---|---|---|---|---|---|
| | Acc | Ma-P | Ma-R | Ma-F | Acc | Ma-P | Ma-R | Ma-F |
| CNN (LeCun et al., 1989) | 74.91 | 74.00 | 78.12 | 73.97 | 87.52 | 88.23 | 88.31 | 88.17 |
| BERT (Devlin et al., 2019) | 80.50 | 80.34 | 81.09 | 78.36 | 89.10 | 90.10 | 89.48 | 89.63 |
| Roberta (Liu et al., 2019) | 79.26 | 78.93 | 81.25 | 78.18 | 90.30 | 91.02 | 90.97 | 90.94 |
| TopJudge (Zhong et al., 2018) | 76.67 | 74.00 | 77.40 | 74.62 | 87.31 | 88.68 | 87.84 | 88.20 |
| R-Former (Dong and Niu, 2021) | 90.71 | **93.06** | 88.66 | **89.82** | 91.54 | 91.61 | **91.96** | **91.58** |
| LADAN (Xu et al., 2020) | 79.64 | 48.43 | 44.13 | 46.18 | 88.09 | 90.12 | 88.82 | 89.47 |
| NeurJudge (Yue et al., 2021) | 71.85 | 69.37 | 71.09 | 68.66 | 82.13 | 82.71 | 82.30 | 82.36 |
| EPM(Feng et al., 2022) | 83.49 | 80.36 | 83.29 | 81.87 | 91.20 | 90.81 | 89.99 | 90.46 |
| CTM(Liu et al., 2022) | 79.33 | 82.39 | 83.12 | 82.81 | 90.28 | 90.34 | 88.08 | 86.30 |
| Dav003 | 44.65 | 52.43 | 32.93 | 35.29 | 25.85 | 35.37 | 25.09 | 22.08 |
| 3.5turbo | 58.37 | 56.03 | 40.68 | 42.62 | 49.65 | 42.29 | 34.05 | 31.85 |
| PLJP(CNN) | 91.62 | 83.43 | 84.88 | 83.40 | 91.49 | 81.80 | 83.95 | 80.06 |
| PLJP(BERT) | **94.18** | 90.25 | **88.67** | 89.05 | **94.99** | **92.12** | 91.10 | 91.33 |

Table 3: Results of charge prediction, the best is **bolded** and the second best is underlined.

includes a fact description accompanied by a complete judgment encompassing three labels: law articles, charges, and prison terms[2].

To mitigate the potential data leakage during the training of LLMs, which were trained on corpora collected until September 2021, we have compiled a new dataset called CJO22. This dataset exclusively contains legal cases that occurred after 2022, sourced from the same origin as CAIL2018[3]. However, due to its limited size, the newly collected CJO22 dataset is inadequate for the training purposes of the domain models. Consequently, we utilize it solely as an additional test set. To facilitate meaningful comparisons, we retain only the labels that are common to both datasets, considering that the labels may not be entirely aligned.

Tab. 1 shows the statistics of the processed datasets, and all the experiments are conducted on the same datasets. For CAIL2018 dataset, we randomly divide it into training set, validation set and test set according to the ratio of 8: 1: 1.

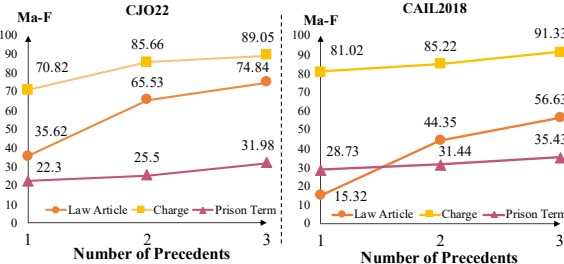

Figure 3: The Ma-F of PLJP with different number of precedents.

The previous cases in the case database are sampled from the training dataset, and we set the amount to 4000.

## 5.2 Baselines

For domain-specific LJP baselines, we implement the following for comparison:

**CNN** (LeCun et al., 1989) extracts text features through convolutional operations with different kernels for text classification; **BERT**(Devlin et al., 2019) is a pre-trained language model and can be easily fine-tuned on the downstream tasks; **Top-Judge** (Zhong et al., 2018) use multi-task learn-

[2]Prison terms are divided into non-overlapping intervals.
[3]https://wenshu.court.gov.cn/

| Method | CJO22 | | | | CAIL2018 | | | |
|---|---|---|---|---|---|---|---|---|
| | Acc | Ma-P | Ma-R | Ma-F | Acc | Ma-P | Ma-R | Ma-F |
| CNN (LeCun et al., 1989) | 27.38 | 18.48 | 17.51 | 17.44 | 34.42 | 32.22 | 30.53 | 31.05 |
| BERT (Devlin et al., 2019) | 36.80 | 29.83 | 27.50 | 27.03 | 40.00 | 37.53 | 33.66 | 33.58 |
| Roberta (Liu et al., 2019) | 29.74 | 24.73 | 24.76 | 23.22 | 40.84 | 38.62 | **38.55** | **38.50** |
| TopJudge (Zhong et al., 2018) | 27.14 | 19.76 | 17.69 | 17.94 | 35.54 | 33.55 | 31.08 | 32.00 |
| R-Former (Dong and Niu, 2021) | 38.63 | 32.63 | 32.76 | 29.51 | 40.70 | 36.09 | 36.76 | 35.04 |
| LADAN (Xu et al., 2020) | 33.69 | 26.40 | 22.94 | 24.55 | 38.03 | 33.66 | 30.08 | 31.77 |
| NeurJudge (Yue et al., 2021) | 26.80 | 26.81 | 26.85 | 25.97 | 33.53 | 36.46 | 37.26 | 36.53 |
| EPM(Feng et al., 2022) | 36.91 | 30.65 | 31.61 | 30.20 | 40.25 | 37.96 | 37.00 | 37.34 |
| CTM(Liu et al., 2022) | 36.81 | 27.10 | 25.96 | 26.46 | 39.56 | 38.66 | 38.02 | 37.84 |
| Dav003 | 0.47 | 5.56 | 0.21 | 0.41 | 0.68 | 10.38 | 0.49 | 0.94 |
| 3.5turbo | 1.40 | 1.16 | 1.07 | 1.11 | 1.02 | 2.71 | 1.13 | 1.15 |
| PLJP(CNN) | 36.51 | 20.21 | 21.44 | 20.07 | 40.81 | 32.77 | 35.59 | 25.71 |
| PLJP(BERT) | **43.52** | **33.37** | **35.67** | **31.98** | **48.72** | **42.64** | 36.80 | 35.43 |

Table 4: Results of prison term prediction, the best is **bolded** and the second best is underlined.

| Method | CJO22 | | | | | | CAIL2018 | | | | | |
|---|---|---|---|---|---|---|---|---|---|---|---|---|
| | Law Article | | Charge | | Prison Term | | Law Article | | Charge | | Prison Term | |
| | Acc | Ma-F | Acc | Ma-F | Acc | Ma-F | Acc | Ma-F | Acc | Ma-F | Acc | Ma-F |
| w/o p | 54.65 | 28.32 | 83.48 | 76.33 | 35.81 | 20.84 | 85.03 | 51.54 | 85.03 | 70.07 | 32.31 | 22.58 |
| w/o c | 45.34 | 40.22 | 42.32 | 41.85 | 32.55 | 20.26 | 67.35 | 46.65 | 72.79 | 60.34 | 26.53 | 13.66 |
| w/o d | **94.18** | **74.84** | 85.58 | 70.50 | 39.53 | 20.31 | 87.07 | 56.63 | 87.41 | 73.45 | 38.09 | 21.44 |
| w/o r | 88.13 | 58.75 | 87.67 | 74.83 | 36.27 | 23.70 | 86.05 | 58.26 | 86.73 | 77.53 | 38.10 | 21.70 |
| w/ e | 90.70 | 67.90 | 80.70 | 66.53 | 35.35 | 20.21 | **89.80** | **61.64** | 85.37 | 68.48 | 38.44 | 23.14 |
| PLJP | **94.18** | **74.84** | 94.18 | 89.05 | **43.52** | **31.98** | 87.07 | 56.63 | **94.99** | **91.33** | **48.72** | **35.43** |

Table 5: Results of ablation experiments, the best is **bolded** and the second best is underlined.

ing and capture the dependencies among the three sub-task in LJP; **NeurJudge** (Yue et al., 2021) splits the fact description into different parts for making predictions; **R-Former** (Dong and Niu, 2021) formalizes LJP as a node classification problem over a global consistency graph and relational learning is introduced; **LADAN** (Xu et al., 2020) uses graph distillation to extract discriminative features of the fact **Retri-BERT** (Chalkidis and Kementchedjhieva, 2023) retrieves similar documents to augment the input document representation for multi-label text classification; **EPM** (Feng et al., 2022) locates event-related information essential for judgment while utilizing cross-task consistency constraints among the subtasks; **CTM** (Liu et al., 2022) establishes a LJP framework with case triple modeling from contrastive case relations.

We use the LLM baselines as follows[4]: **Dav003** means the text-davinci-003, **3.5turbo** means the gpt-3.5-turbo. These LLMs are both from the GPT-3.5 family, released by OpenAI and can understand and generate complex natural language[5].

For PLJP, we take the CNN and BERT as the predictive models, and take the text-davinci-003 as the implementation of the LLM, named as PLJP(CNN)

and PLJP(BERT). The top-k accuracy of CNN and BERT is shown in the Appendix. Considering the length limit of the prompt, we set the number of precedents to 3.

We also do ablation experiments as follows: **PLJP w/o p** refers to the removal of precedents, and the prediction of labels is done solely based on the candidate labels using the LLM; **PLJP w/o c** denotes we remove the candidate labels and predict the label only with the fact description and precedents; **PLJP w/o d** means we predict the three labels independently instead of considering the dependencies among the three subtasks; **PLJP w/o r** denotes we find precedents based the raw fact instead of from the reorganized fact; **PLJP w/ e** means we let the LLMs generate the explanation of the prediction as well.

In the ablation study, PLJP means PLJP(BERT).

### 5.3 Experiment Settings

Here we describe the implementation of PLJP in our experiments. Note all the LLMs and domain models are replaceable in the PLJP framework.

In the experiments, for the LLMs, we directly use the APIs provided by OpenAI. For the domain models, we use the unsupervised dense retrieval model (Izacard et al., 2022) in precedent retrieval, which gets the precedents from the case database

---

[4]We give a fixed example in the prompt to help the LLMs understand the tasks.

[5]https://platform.openai.com/docs/models

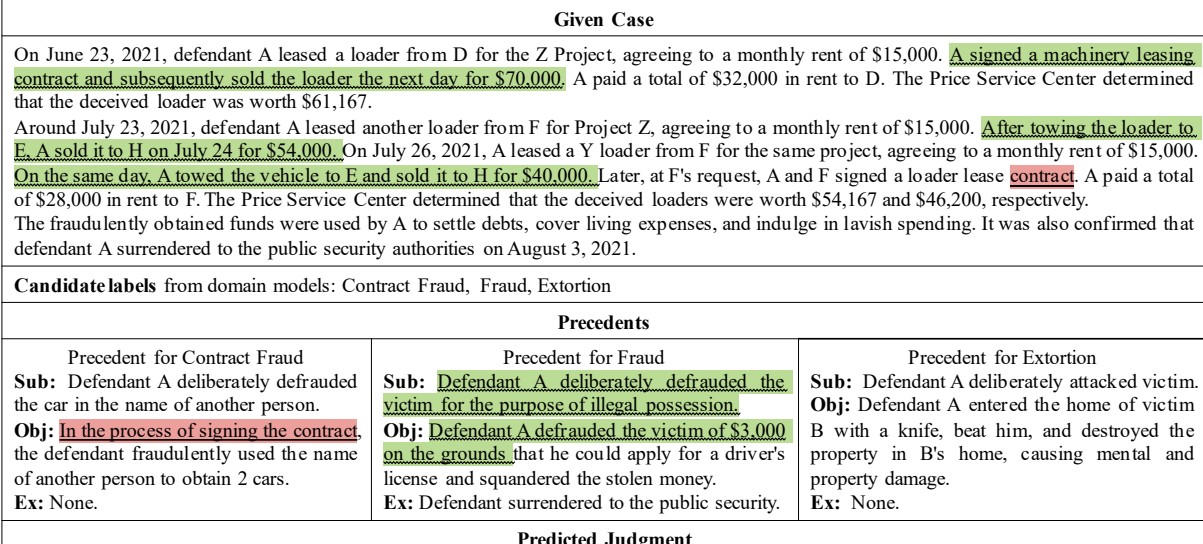

| **Given Case** |
| On June 23, 2021, defendant A leased a loader from D for the Z Project, agreeing to a monthly rent of $15,000. A signed a machinery leasing contract and subsequently sold the loader the next day for $70,000. A paid a total of $32,000 in rent to D. The Price Service Center determined that the deceived loader was worth $61,167. |

**Given Case**

On June 23, 2021, defendant A leased a loader from D for the Z Project, agreeing to a monthly rent of $15,000. A signed a machinery leasing contract and subsequently sold the loader the next day for $70,000. A paid a total of $32,000 in rent to D. The Price Service Center determined that the deceived loader was worth $61,167.

Around July 23, 2021, defendant A leased another loader from F for Project Z, agreeing to a monthly rent of $15,000. After towing the loader to E, A sold it to H on July 24 for $54,000. On July 26, 2021, A leased a Y loader from F for the same project, agreeing to a monthly rent of $15,000. On the same day, A towed the vehicle to E and sold it to H for $40,000. Later, at F's request, A and F signed a loader lease contract. A paid a total of $28,000 in rent to F. The Price Service Center determined that the deceived loaders were worth $54,167 and $46,200, respectively.

The fraudulently obtained funds were used by A to settle debts, cover living expenses, and indulge in lavish spending. It was also confirmed that defendant A surrendered to the public security authorities on August 3, 2021.

**Candidate labels** from domain models: Contract Fraud, Fraud, Extortion

**Precedents**

| Precedent for Contract Fraud | Precedent for Fraud | Precedent for Extortion |
|---|---|---|
| **Sub:** Defendant A deliberately defrauded the car in the name of another person. **Obj:** In the process of signing the contract, the defendant fraudulently used the name of another person to obtain 2 cars. **Ex:** None. | **Sub:** Defendant A deliberately defrauded the victim for the purpose of illegal possession. **Obj:** Defendant A defrauded the victim of $3,000 on the grounds that he could apply for a driver's license and squandered the stolen money. **Ex:** Defendant surrendered to the public security. | **Sub:** Defendant A deliberately attacked victim. **Obj:** Defendant A entered the home of victim B with a knife, beat him, and destroyed the property in B's home, causing mental and property damage. **Ex:** None. |

**Predicted Judgment**

| R-Former: Contract Fraud ✗ | BERT: Contract Fraud ✗ | PLJP(BERT): Fraud ✅ |
|---|---|---|

Figure 4: The charge prediction of a given case. The green parts are useful information for prediction, while the red parts are content that can be confused by the domain models.

according to the reorganized facts. For other domain models such as TopJudge and NeurJudge, we use the training settings from the original paper.

For the metrics, we employ Accuracy (Acc), Macro-Precision (Ma-P), Macro-Recall (Ma-R) and Macro-F1 (Ma-F).

## 5.4 Experiment Results

We analyze the experimental results in this section.

**Result of judgment prediction:** From Tab. 2, Tab. 3 and Tab. 4, we have the following observations: 1) The LLMs perform not well in the prediction tasks alone, especially when the label has no actual meaning (e.g., the index of the law article and prison term). 2) By applying our PLJP framework with the collaboration of LLMs and domain models, the simple models (e.g., CNN, BERT) gain significant improvement. 3) The model performance on CJO22 is lower than that on CAIL2018, which shows the challenge of the newly constructed test set. 4) PLJP(BERT) achieves the best performance in almost all the metric evaluation metrics in both CAIL2018 and CJO22 test sets, which proves the effectiveness of the PLJP. 5) Compared to the prediction of the law article and charge, the prediction of prison term is still a more challenging task. 6) The reported results of the LJP baselines are not as good as the original papers, this may be because we keep all the low-frequency labels instead of removing them as the original papers did.

**Results of ablation experiment:** From Tab. 5, we can conclude that: 1) The performance gap of the PLJP w/o p and PLJP demonstrates the effects of the precedents. 2) The results of PLJP w/o c prove the importance of the candidate labels. 3) Considering the topological dependence of the three sub-tasks benefits the model performance as PLJP w/o d shows. 4) When we use the raw fact instead of the reorganized fact, the performance drops (e.g., the Acc of prison term in CJO22 drops from 45.32% to 36.27%). 5) If we force the LLMs to generate the explanation of the prediction, the performance also drops a bit. We put cases with explanations in the Appendix.

From Fig. 3, we can find that the performance of PLJP improves as the number of precedents increases, which also proves the effectiveness of injecting precedents into the LJP.

## 5.5 Case Study

Fig. 4 shows an intuitive comparison among the three methods in the process of charge prediction. Based on the fact description of the given case, the domain models provide candidate charges with the corresponding precedents. As the case shows, the defendant made fraud by selling the cars that were rented from other people. However, since there contains "contract" in the fact description, baselines (e.g., R-Former and BERT) can be misled and predict the wrong charge of "Contract Fraud". Through an in-context precedent comprehension by the LLMs, PLJP(BERT) distinguishes the dif-

ferences among the precedents and the given case (e.g., the crime does not occur during the contracting process, and the contract is only a means to commit the crime), and give the right result of "Fraud".

# 6 Conclusion and Future Work

In this paper, we address the important task of legal judgment prediction (LJP) by taking precedents into consideration. We propose a novel framework called precedent-enhanced legal judgment prediction (PLJP), which combines the strength of both LLMs and domain models to better utilize (e.g., retrieve and comprehend) the precedents. Experiments on the real-world dataset prove the effectiveness of the PLJP.

Based on the PLJP, in the future, we can explore the following directions: 1) Develop methods to identify and mitigate any biases that could affect the predictions and ensure fair and equitable outcomes. 2) Validate the effectiveness of LLM and domain collaboration in other vertical domains such as medicine and education.

## 6.1 Ethical Discussion

With the increasing adoption of Legal AI in the field of legal justice, there has been a growing awareness of the ethical implications involved. The potential for even minor errors or biases in AI-powered systems can lead to significant consequences.

In light of these concerns, we have to claim that our work is an algorithmic exploration and will not be directly used in court so far. Our goal is to provide suggestions to judges rather than making final judgments without human intervention. In practical use, human judges should be the final safeguard to protect justice fairness. In the future, we plan to study how to identify and mitigate potential biases to ensure the fairness of the model.

# 7 Limitations

In this section, we discuss the limitations of our works as follow:

• We only interact with the LLMs one round per time. The LLMs are capable of multi-round interaction (e.g., Though of Chains), which may help the LLM to better understand the LJP task.

• We validate the effectiveness of LLM and domain model collaboration in the legal domain. It's worthwhile to explore such collaboration in other vertical domains such as medicine and education,

as well as in other legal datasets (e.g., the datasets from the Common Law system).

# Acknowledgments

This work was supported in part by National Key Research and Development Program of China (2022YFC3340900), National Natural Science Foundation of China (62037001, 62376243, U20A20387), the StarryNight Science Fund of Zhejiang University Shanghai Institute for Advanced Study (SN-ZJU-SIAS-0010), Young Elite Scientists Sponsorship Program by CAST (2021QNRC001), Project by Shanghai AI Laboratory (P22KS00111), Program of Zhejiang Province Science and Technology (2022C01044), Fundamental Research Funds for the Central Universities (226-2023-00060), Key Research and Development Program of Zhejiang Province (2021C01013).

Finally, we would like to thank the anonymous reviewers for their helpful feedback and suggestions.

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

# A   Appendices

## A.1   Top-k Accuracy

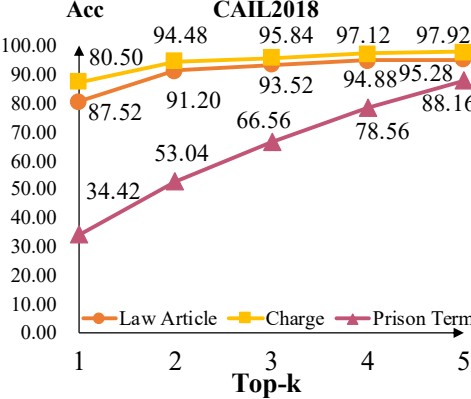

Figure 5: The top-k accuracy of CNN on CAIL dataset.

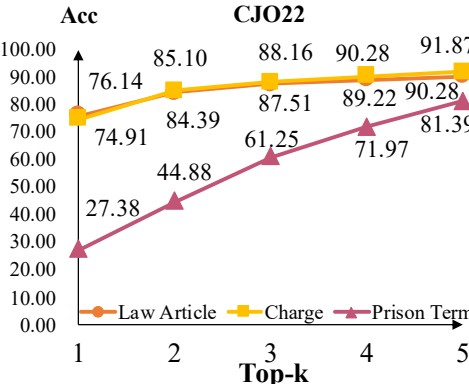

Figure 6: The top-k accuracy of CNN on CJO22 dataset.

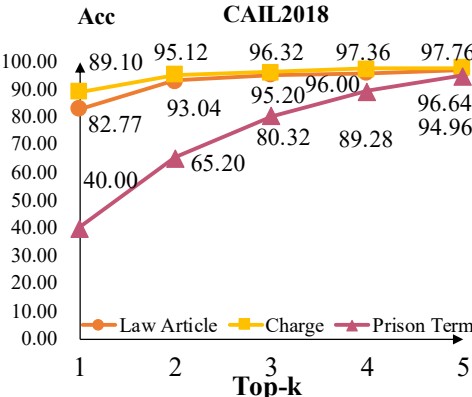

Figure 7: The top-k accuracy of BERT on CAIL dataset.

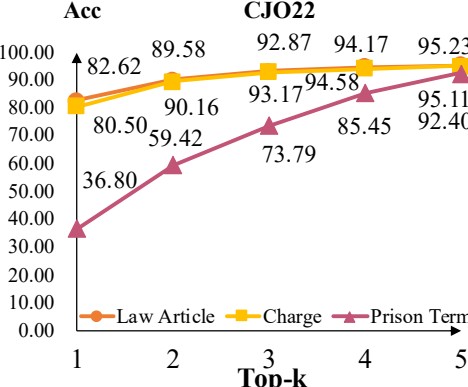

Figure 8: The top-k accuracy of BERT on CJO22 dataset.

## A.2 More Show Cases

| Given Case |
|---|
| At about 19:00 on September 12, 2021, when defendant A was driving a two-wheeled motorcycle along National Highway 106 from south to north to the road in front of Hotel C, he hit pedestrian B who was crossing the road from east to west in front, causing B severe head injury, and later B died after ineffective treatment. The D County Traffic Police Brigade determined that A was primarily responsible for the accident. After the case, defendant A immediately reported the case and waited at the scene for the traffic police to come and deal with it. Later, defendant A reached a compensation agreement with the victim's relatives, compensating the victim's relatives for economic losses of $260,000 in one lump sum, and obtained the understanding of the victim's relatives. |
| Candidate labels from domain models: Causing Traffic Casualties, Dangerous Driving, Involuntary Manslaughter |

| **Precedents** | | |
|---|---|---|
| Precedent for Causing Traffic Casualties | Precedent for Dangerous Driving | Precedent for Involuntary Manslaughter |
| **Sub:** Defendant A drove a decked two-wheeled motorcycle without a license after drinking alcohol, and failed to drive safely in accordance with the operating specifications, resulting in the death of a pedestrian. **Obj:** Defendant A drove a two-wheeled motorcycle under the influence of alcohol, drove east to west along the X route of C City to the section of Z Village in Y Town, C City, and injured pedestrian B walking in the same direction and died after ineffective rescue. **Ex:** None. | **Sub:** Defendant A was driving a gray Chevrolet minibus under the influence of alcohol and collided. **Obj:** Defendant A's blood alcohol content was 151.4mg/100ml, and he was seized by the police on the spot. **Ex:** After the case, defendant A and B reached a mediation agreement on civil compensation. | **Sub:** Defendant A drove a two-wheeled motorcycle along National Highway 206 from north to south and hit pedestrian B. **Obj:** Defendant A knocked B to the ground, and then A sent B to the hospital. B died after ineffective rescue, and after forensic physical examination, B died of severe head injury. **Ex:** Defendant A voluntarily surrendered to the C Branch of the D City Public Security Bureau, and his punishment can be mitigated according to law. |

| **Predicted Judgment** | | |
|---|---|---|
| R-Former | BERT | PLJP(BERT) |
| Dangerous Driving ❌ | Dangerous Driving ❌ | Causing Traffic Casualties ✅ |

Figure 9: More case 1.

| Given Case |
|---|
| Around January 2021, without obtaining a forest harvesting permit, defendant A hired personnel to harvest the trees located in his own mountain farm next to the "Great Waterfall" in B Village, C Town, D County, and used the felled trees to build houses and sell them. Among them, the total profit from the sale of felled trees was about $2,000. A total of 14.149 cubic meters of trees were identified as being harvested. On November 9, 2021, defendant A voluntarily surrendered to the E Police Station of D County after being notified by the police handling the case. |
| Candidate labels from domain models: Illegal Denudation, Illegal Lumbering, Illegal Occupation of Agricultural Land |

| **Precedents** | | |
|---|---|---|
| Precedent for Illegal Denudation | Precedent for Illegal Lumbering | Precedent for Illegal Occupation of Agricultural Land |
| **Sub:** Defendant A cut down trees protected by the state without the approval of the forestry authorities. **Obj:** Defendant A did not apply for a forest harvesting permit, and cut down trees in the "Beofu Mountain" of in B Village, C Town, with a total area of 4.5 acres of trees cut down and a total of 20.8702 cubic meters of standing wood accumulation. **Ex:** Defendant A voluntarily surrenders after committing a crime and truthfully confesses his crime, which is a voluntary surrender, and the punishment can be mitigated according to law. | **Sub:** Defendant A intentionally committed the crime for the purpose of illegal possession. **Obj:** Defendant A falsely claimed that 328 poplar trees located in the northeast of X Village, Y Town, Z County, owned by himself, sold the above-mentioned poplars to E, and then felled them at a price of $6,600, with a standing log accumulation of 15.7987 cubic meters and a total value of $12,068. **Ex:** None. | **Sub:** Defendant A illegally reclaimed forest land without obtaining legal formalities. **Obj:** Defendant A illegally reclaimed 10.05 acres of forest land in the Willow River Forest Farm Application Area of the B Forestry Bureau. **Ex:** Defendant A's confession and defense and evidence such as the on-site investigation records of the B Branch of the C Public Security Bureau confirmed that it was recommended that the punishment be mitigated according to law. |

| **Predicted Judgment** | | |
|---|---|---|
| R-Former | BERT | PLJP |
| Illegal Denudation ✅ | Illegal Lumbering ❌ | Illegal Denudation ✅ |

Figure 10: More case 2.

| Given Case |
|---|
| At about 18:27 on December 1, 2021, defendant A drove a small ordinary bus from west to east along Wenhua Road, Qingshan District, Baotou City to a point 150 meters west of the intersection of Wenhua Road and Yingbin Road (the east side of the front of Qingshan Hotel) pedestrian crossing) and collided with the victim B who was riding an electric bicycle across the road from north to south, resulting in the death of B on the spot. After the accident, defendant A dialed 120 emergency number and 122 alarm number to call the police, and waited for the arrival of the traffic police at the scene. According to the forensic autopsy examination and identification of Baotou Public Security Traffic Judicial Appraisal Center, B died of severe craniocerebral injury caused by a traffic accident. According to the determination of the road traffic accident by the Qingshan Brigade of the Baotou Traffic Management Detachment, the defendant A bears full responsibility for the accident, and B has no responsibility for the accident. It was also found that the small ordinary passenger car that caused the accident had purchased compulsory motor vehicle traffic accident liability insurance and motor vehicle third party liability insurance at Baotou City Center Sub-branch of Inner Mongolia Branch of a certain property insurance company, and the accident occurred within the insurance period. . It was also found out that after the accident, defendant A had paid $40,000 for the funeral expenses of the family members of the victim. It was further found that defendant A voluntarily prepaid compensation of $10,000 to this court during the trial period. |

**Predicted Judgment and Explanation**

| Law Article | Charge | Prison Term |
|---|---|---|
| Law article truth: 133
Law article prediction: Article 133. Because of the facts of the case enclosed by <>, defendant A drove a small ordinary bus along Wenhua Road from west to east to 150 meters west of the intersection of Wenhua Road and Yingbin Road, and met the victim B who was riding an electric bicycle across the road from north to south. If two people collided and violated traffic management regulations, thereby causing injury or death to the victim, it constitutes the crime of causing a traffic accident according to law, so the corresponding legal article in this case should be Article 133. | Charge truth: Causing traffic casualties
Charge prediction: Causing traffic accident. Since the three types of cases all show that the defendant did not drive the vehicle in accordance with traffic safety regulations, thus causing a traffic accident and directly causing the death of the victim on the spot, the crime of causing a traffic accident can be selected in this case, which is in line with Article 133 of the "Criminal Law of the People's Republic of China". Article 133 stipulates: "Whoever drives a motor vehicle carelessly and the circumstances are serious shall be sentenced to fixed-term imprisonment of not more than three years or criminal detention, and shall also be fined." | Prison term truth:17 months
Prison term prediction: One to two years in prison. Since the driver in this case did not follow the traffic rules and lost his correct judgment, the victim died on the spot, and the crime was serious, so a longer sentence should be imposed, so he chose a fixed-term imprisonment of one to two years. Precedents also support the adoption of a fixed-term imprisonment of one to two years in this case. The crimes of two of the three precedents are all lighter than this case, and they are all sentenced to a correspondingly shorter sentence, which proves that a longer sentence should be adopted in this case to achieve the disciplinary effect. |

Figure 11: Case with explanations 1.

| Given Case |
|---|
| In the early morning of March 11, 2022, defendant A drove a car carrying B, C, D and others in the county seat, and then defendant A left B, C, and D in the car he was driving to smoke the drug ketamine on the back seat. At about 10:00 on March 18, 2022, defendant A was arrested at a milk tea shop in Rucheng County, and later truthfully confessed the above facts. |

**Predicted Judgment and Explanation**

| Law Article | Charge | Prison Term |
|---|---|---|
| Law article truth: 354
Law article prediction: Article 354. Compared with precedents, the subjective motive of this case is to allow others to take drugs, and the objective behavior is to takes the drug ketamine. The behaviors involved are all "allowing others to take drugs", so the applicable legal provisions For Article 354, the reason is that the law involves the penalties for drug detention. The difference from this case is that precedents involved drunk driving (Article 133) or mob fighting (Article 292), and the extraneous circumstances were not exactly the same, so the applicable laws and regulations were different from this case. | Charge truth: Providing venues for drug users
Charge prediction: Providing venues for drug users. Comparing the three aspects of subjective motivation, objective behavior, and incidental circumstances, it can be seen that defendant A's subjective motivation was intentional, and his objective behavior was to allow others to take drugs. The circumstances outside the incident are different. In this case, defendant A can be given a lighter punishment according to law after being brought to justice. From this, it can be judged that the defendant in this case was charged with allowing others to take drugs. | Prison term truth: 7 months
Prison term prediction: Six to nine months in prison. The objective behavior of this case is clear. The vehicle driven by defendant A carried poisonous substances; in addition, the subjective motive and extraneous circumstances of this case are more in line with the original intention of a lighter punishment than precedents. After being arrested and brought to justice, defendant A truthfully confessed his crime. Therefore, the sentence in this case was chosen as fixed-term imprisonment of six to nine months. |

Figure 12: Case with explanations 2.

| Given Case |
|---|
| In March 2019, when defendant A and victim B dated in the name of boyfriend and girlfriend, from March to September 2019, A demanded money from B on the grounds of helping B to transfer work and handling matters such as customs seizure , B delivered a total of $129,028 to A through WeChat, Alipay transfer and cash payment, and most of the money was used by A for personal squandering. When B asked for repayment, A shirked on the grounds of being hospitalized and deactivating the account. In April 2019, defendant A lied to victim C that he could help deal with C's husband Shi Moumou's drunk driving, and demanded money from C in the name of dredging the relationship. C transfers a total of $9,000 to A through WeChat, which is used by A for personal consumption. When C asked for repayment, A forged the record of repayment to C's WeChat and blacklisted C's WeChat. On March 5, 2020, A returned $9,000 to the victim C. It was also found that because defendant A suffered from a serious illness, the Rencheng District People's Court of Jining City sentenced defendant A to six months' imprisonment, which has not yet been executed. It was further found that on March 15, 2022, defendant A paid victim B $120,000 in cash and issued a bill signed in acknowledgement of debt, which recorded "A voluntarily paid victim B $129,000 , and paid $120,000 in cash first, and owed $9,000 was paid off in four months, $2,000 was paid every month for the first three months, and the last $3000 was paid off before July 15, 2022." Victim B issued a letter of understanding, expressing his respect for A's behavior. Forgive me and agree to a lighter punishment. Defendant A voluntarily pleaded guilty and accepted punishment, and had no objection to the facts, charges and sentencing recommendations of the public prosecution agency. The defender has no objection to the alleged criminal facts and charges, but believes that defendant A confessed the facts of the case before taking compulsory measures, which constitutes surrender and confession; A's fraudulent behavior occurred in the process of dating, and the subjective viciousness is relatively small and the harm is not great ; The amount of suspected fraud is not clear. According to the principle of lenient crime, the statutory penalty of "huge amount" should not be applied. It is recommended that the defendant be exempted from criminal punishment under the circumstances of surrendering himself, actively compensating the victim and obtaining forgiveness. |

| Predicted Judgment and Explanation | | |
|---|---|---|
| Law Article | Charge | Prison Term |
| Law article truth: 266 Law article prediction: Article 266, because the defendant in this case took the means of fraudulently using another person's account number, fictitious facts and concealing the truth to obtain other people's property in a large amount, the provisions of the said article 266 apply. In the content of the precedents, in the first case, the defendant used the means of proposing fictitious facts to cheat money, in the second case, the defendant used the fraudulent use of other people's accounts credit card to demand money owed, and in the third case, the defendant used the means of fraudulent use of other people's accounts to cheat, so this case and the three precedents all involve cheating property, and the acts in the three cases are mainly fraud, so the provisions of article 266 are also applicable to this case. | Charge truth: Fraud Charge prediction: Fraud. Because both this case and the precedents were established by the defendant with the purpose of unlawful possession, the act of taking possession of money by fictitious facts and concealing the truth are fraudulent. This case and the precedents both take the apparently normal means, such as using credit cards to obtain property, both constitute fraud. | Prison term truth: 36 months. Prison term prediction: Two to three years in prison. From the above three categories of cases, it can be seen that the subjective motive of the defendant as well as the objective behavior is basically the same as this case, and the extraneous circumstances of the defendant in this case are also more prominent than them. The defendant recognized the amount of fraud, truthfully confessed to the crime, cooperated with the investigation and confessed the facts, which can be used as mitigating factors in the sentence, so the sentence of the defendant in this case is chosen to be two to three years in prison, and the other two categories of cases also have the same sentence. However, fine adjustments should be made according to the actual circumstances. |

Figure 13: Case with explanations 3.