# OpenReview forum: "Precedent-Enhanced Legal Judgment Prediction with LLM and Domain-Model Collaboration"
_EMNLP/2023/Conference — EMNLP 2023 Main_

### Official Review · Reviewer_ZR8e · 2023-07-23

**Soundness:** 2

**Excitement:**

4: Strong: This paper deepens the understanding of some phenomenon or lowers the barriers to an existing research direction.

**Paper Topic And Main Contributions:**

In this paper, the authors propose a precedent-enhanced LJP framework called PLJP, which leverages the strength of both LLM and domain models in the context of precedents. Experiments on the real-world dataset demonstrate the effectiveness of PLJP.

**Questions For The Authors:**

Please see the Reasons To Reject.

**Reasons To Accept:**

1. This paper seems the first to utilize the precedents to enhance the performance of LJP tasks.
2. This paper uses LLM reasonably.
3. Extensive experiments demonstrate the effectiveness of the proposed method.

**Reasons To Reject:**

The writing of this paper is not clear. In section 4.2.1, the authors said that they chose the case that has the highest similarity score and has the same label as the precedents. However, since the final task is to predict the label of charge, law articles, etc., how to get the label before that? Moreover, it is not clear how they utilize the precedent "serves as a supplemental explanation of the label".



**Reproducibility:**

4: Could mostly reproduce the results, but there may be some variation because of sample variance or minor variations in their interpretation of the protocol or method.

**Reviewer Confidence:**

5: Positive that my evaluation is correct. I read the paper very carefully and I am very familiar with related work.

---

> ### Author Rebuttal · Authors · 2023-08-29
>
> We are grateful for your thoughtful input and dedication to this paper. Our replies can be found below.
>
> Q1-1:
>
>     For the concern about "label in the precedent selection process":
>
>
> - We would like to highlight that the label we’re referring to at this stage is the "candidate label", not the "final label".
> - As described in section 4.2.1, for each candidate label generated by the predictive model (Line 216), we retrieve one case that has the same label from the case database as the precedent, according to the similarity scores calculated by the retrieval model (Line 317).
> - If there are multiple candidate labels, we retrieve corresponding precedents for each of them. It is essential to note that a single case might be accompanied by a variety of actualized precedents, depending on the underlying task; for instance, in the context of law article prediction, charge prediction and prison term prediction.
>
> Q1-2:
>
>     For the concern about "utilization of the precedent serves as a supplemental explanation of the label":
>
> - Since the precedents are selected by the candidate labels, it is possible to view each precedent as an example of the corresponding candidate label, which contributes to the in-context learning of the LLM in the final prediction.
>
> We will revise the writing and ensure the clarity of the paper.

---

### Official Review · Reviewer_GcEk · 2023-07-31

**Soundness:** 4

**Excitement:**

4: Strong: This paper deepens the understanding of some phenomenon or lowers the barriers to an existing research direction.

**Missing References:**

- Ilias Chalkidis, Ion Androutsopoulos, and Nikolaos Aletras. 2019. Neural Legal Judgment Prediction in English. In Proceedings of the 57th Annual Meeting of the Association for Computational Linguistics, pages 4317–4323, Florence, Italy. Association for Computational Linguistics.
- Joel Niklaus, Ilias Chalkidis, and Matthias Stürmer. 2021. Swiss-Judgment-Prediction: A Multilingual Legal Judgment Prediction Benchmark. In Proceedings of the Natural Legal Language Processing Workshop 2021, pages 19–35, Punta Cana, Dominican Republic. Association for Computational Linguistics.
- Ilias Chalkidis, Abhik Jana, Dirk Hartung, Michael Bommarito, Ion Androutsopoulos, Daniel Katz, and Nikolaos Aletras. 2022. LexGLUE: A Benchmark Dataset for Legal Language Understanding in English. In Proceedings of the 60th Annual Meeting of the Association for Computational Linguistics (Volume 1: Long Papers), pages 4310–4330, Dublin, Ireland. Association for Computational Linguistics.
- Vijit Malik, Rishabh Sanjay, Shubham Kumar Nigam, Kripabandhu Ghosh, Shouvik Kumar Guha, Arnab Bhattacharya, and Ashutosh Modi. 2021. ILDC for CJPE: Indian Legal Documents Corpus for Court Judgment Prediction and Explanation. In Proceedings of the 59th Annual Meeting of the Association for Computational Linguistics and the 11th International Joint Conference on Natural Language Processing (Volume 1: Long Papers), pages 4046–4062, Online. Association for Computational Linguistics.
- Ilias Chalkidis* and Yova Kementchedjhieva*. Retrieval-augmented Multi-label Text Classification. arXiv. 2023.

**Paper Topic And Main Contributions:**

The authors propose a new pipelined framework for Legal Judgment Prediction (LJP) for the Chinese CAIL multi-task datasets, where a model has to predict the relevant criminal charge, law articles, and penalty term based on criminal case facts. The new framework has the following steps (modules): (a) Fact summarization using an LLM, e.g., ChatGPT, (b) Judgment prediction/classification with a domain/task-specific classifier, (c) Relevant case law (precedent) retrieval with a domain-specific neural embedder, and (d) in-context LJP by an LLM using as inputs the summarized facts of the given case, and k retrieved relevant precedent cases, and the prior prediction from step b. The authors compare the new framework with several prior-art (baselines) in the CAIL2018 dataset and a collection of new cases from 2022, dubbed CJO22.

The new framework has improved results in penalty term, and law article prediction, and comparable performance in charge prediction compared to the best prior-art models (Ar-Former, NeurJudge). The authors perform ablations un-plugging different steps (modules) from the full pipeline, e.g., w/o summarization, or w/o retrieval-augmentation, which show the importance of the selected steps (modules).

**Questions For The Authors:**

I would be interested to hear the authors' thoughts on my "negative" comments (reasons to reject).

**Reasons To Accept:**

- The paper proposes an interesting retrieval-augmented approach utilizing LLMs for the task of LJP.

**Reasons To Reject:**

- There are no other retrieval-augmented BERT-based (prior-art) baselines utilizing the same or similar steps with the proposed framework. Prior-art models enhanced with retrieval-augmentation to condition on precedent (summarized) cases, may perform equally well to LLMs.
- The proposed method is heavily dependent on a fine-tuned BERT-based classifier. In other words, it can be understood as a few-shot (precedent cases) system that also considers predictions for the new input from a strong baseline.

**Reproducibility:**

3: Could reproduce the results with some difficulty. The settings of parameters are underspecified or subjectively determined; the training/evaluation data are not widely available.

**Reviewer Confidence:**

5: Positive that my evaluation is correct. I read the paper very carefully and I am very familiar with related work.

**Typos Grammar Style And Presentation Improvements:**

- line 54: "courses" --> sources
- The authors should make clear and mention that they use datasets for LJP in the Chinese language in the context of People's Republic of China (PRC) jursdiction (law). There are phrases like "The legal judgment typically includes the law article, charge and prison term.", which sound like the LJP is a universal task across languages and countries. The Bender Rule: "Always name the language you're working on" is not only applicable to English.

---

> ### Author Rebuttal · Authors · 2023-08-29
>
> Thank you for your insightful feedback. We are committed to improving our research study, so we eagerly look forward to incorporating your suggestions for further refinement. Please find our responses in the following content.
>
> Q1:
>
>     For the concern about "retrieval-augmented BERT-based baseline":
>
> - Thanks for the suggestion. We added a retrieval-augmented BERT-based baseline, named Retrieval BERT. We implement the Retrieval BERT referring to [1], and we set the number of retrieved documents (precedents) to 3, the results are shown below:
>
> #### Task: Law article prediction
>
> #### Dataset: CJO22
>
> | Method         | Acc       | Ma-P      | Ma-R      | Ma-F      |
> | -------------- | --------- | --------- | --------- | --------- |
> | Retrieval BERT | 82.29     | 45.64     | 44.79     | 44.01     |
> | PLJP(CNN)      | 87.67     | 55.21     | 55.59     | 54.37     |
> | PLJP(BERT)     | **94.18** | **74.65** | **76.23** | **78.84** |
>
> #### Dataset: CAIL2018
>
> | Method         | Acc       | Ma-P      | Ma-R      | Ma-F      |
> | -------------- | --------- | --------- | --------- | --------- |
> | Retrieval BERT | 79.00     | 48.38     | 48.28     | 47.60     |
> | PLJP(CNN)      | 86.05     | 58.08     | 56.46     | 54.92     |
> | PLJP(BERT)     | **87.07** | **58.81** | **57.29** | **56.63** |
>
> #### Task: Charge prediction
>
> #### Dataset: CJO22
>
> | Method         | Acc       | Ma-P      | Ma-R      | Ma-F      |
> | -------------- | --------- | --------- | --------- | --------- |
> | Retrieval BERT | 82.11     | 82.89     | 80.45     | 78.00     |
> | PLJP(CNN)      | 91.62     | 83.43     | 84.88     | 83.40     |
> | PLJP(BERT)     | **94.18** | **90.25** | **88.67** | **89.05** |
>
> #### Dataset: CAIL2018
>
> | Method         | Acc       | Ma-P      | Ma-R      | Ma-F      |
> | -------------- | --------- | --------- | --------- | --------- |
> | Retrieval BERT | 92.37     | 91.51     | 90.58     | 90.18     |
> | PLJP(CNN)      | 91.49     | 81.80     | 83.95     | 80.06     |
> | PLJP(BERT)     | **94.99** | **92.12** | **91.10** | **91.33** |
>
> #### Task: Prison term prediction
>
> #### Dataset: CJO22
>
> | Method         | Acc       | Ma-P      | Ma-R      | Ma-F      |
> | -------------- | --------- | --------- | --------- | --------- |
> | Retrieval BERT | 34.93     | 27.36     | 25.10     | 25.89     |
> | PLJP(CNN)      | 36.51     | 20.21     | 21.44     | 20.07     |
> | PLJP(BERT)     | **43.52** | **33.37** | **35.67** | **31.98** |
>
> #### Dataset: CAIL2018
>
> | Method         | Acc       | Ma-P      | Ma-R      | Ma-F      |
> | -------------- | --------- | --------- | --------- | --------- |
> | Retrieval BERT | 39.64     | 35.74     | 34.38     | 33.66     |
> | PLJP(CNN)      | 40.81     | 32.77     | 35.59     | 25.71     |
> | PLJP(BERT)     | **48.72** | **42.64** | **36.80** | **35.43** |
>
> - From the result, we can find that PLJP outperforms the Retrieval BERT consistently in both datasets, this may be because the LLMs have a better ability to comprehend the complex precedents' facts than the BERT model.
> - We will discuss this issue in detail in the revision.
>
> Q2:
>
>     For the concern about "BERT-based classifier is a strong baseline":
>
> - Our method is a collaboration of LLM and domain models such as a fine-tuned BERT classifier. Each part of the collaboration holds significance.
> - By examining the results presented in Tables 2, 3, and 4, we can conclude that:
>   - The efficacy of domain models directly influences the overall performance. For instance, PLJP(BERT) outperforms PLJP(CNN), highlighting that superior domain models lead to improved outcomes.
>   - The collaboration of LLMs and domain-specific models in the PLJP yields performance enhancements. A notable example is the charge prediction accuracy in CJO22, which sees an increase from 80.5% to 94.18% with PLJP(BERT).
>
> - Therefore, the PLJP benefits from both domain models and LLMs. The collaboration between the domain models and LLMs contributes to the overall effectiveness of the approach.
>
>
>
> Typos and missing references:
>
> - Thanks for the reminder, we will make clear the language we are working on in the revision. Also, we will ensure all the missing references are included in the revision [1-5].
>
>
>
> Reference:
>
> [1]Ilias Chalkidis, Ion Androutsopoulos, and Nikolaos Aletras. 2019. Neural Legal Judgment Prediction in English. In Proceedings of the 57th Annual Meeting of the Association for Computational Linguistics, pages 4317–4323, Florence, Italy. Association for Computational Linguistics.
>
> [2]Joel Niklaus, Ilias Chalkidis, and Matthias Stürmer. 2021. Swiss-Judgment-Prediction: A Multilingual Legal Judgment Prediction Benchmark. In Proceedings of the Natural Legal Language Processing Workshop 2021, pages 19–35, Punta Cana, Dominican Republic. Association for Computational Linguistics.
>
> [3]Ilias Chalkidis, Abhik Jana, Dirk Hartung, Michael Bommarito, Ion Androutsopoulos, Daniel Katz, and Nikolaos Aletras. 2022. LexGLUE: A Benchmark Dataset for Legal Language Understanding in English. In Proceedings of the 60th Annual Meeting of the Association for Computational Linguistics (Volume 1: Long Papers), pages 4310–4330, Dublin, Ireland. Association for Computational Linguistics.
>
> [4]Vijit Malik, Rishabh Sanjay, Shubham Kumar Nigam, Kripabandhu Ghosh, Shouvik Kumar Guha, Arnab Bhattacharya, and Ashutosh Modi. 2021. ILDC for CJPE: Indian Legal Documents Corpus for Court Judgment Prediction and Explanation. In Proceedings of the 59th Annual Meeting of the Association for Computational Linguistics and the 11th International Joint Conference on Natural Language Processing (Volume 1: Long Papers), pages 4046–4062, Online. Association for Computational Linguistics.
>
> [5]Ilias Chalkidis* and Yova Kementchedjhieva*. Retrieval-augmented Multi-label Text Classification. arXiv. 2023.

---

### Official Review · Reviewer_9C2g · 2023-08-01

**Soundness:** 3

**Excitement:**

4: Strong: This paper deepens the understanding of some phenomenon or lowers the barriers to an existing research direction.

**Missing References:**

[1] Dugang Liu, Weihao Du, Lei Li, Weike Pan, and Zhong Ming. 2022. Augmenting legal judgment prediction with contrastive case relations.
    In Proceedings of the 29th International Conference on Computational Linguistics, pages 2658–2667, Gyeongju, Republic of Korea.
    International Committee on Computational Linguistics.
[2] Yi Feng, Chuanyi Li, and Vincent Ng. 2022. Legal judgment prediction via event extraction with constraints. In Proceedings of the 60th
    Annual Meeting of the Association for Computational Linguistics (Volume 1: Long Papers), pages 648–664, Dublin, Ireland. Association for
    Computational Linguistics.


**Paper Topic And Main Contributions:**

The topic of this paper is about approaches for data- and compute efficiency.
The main contributions of this paper are as follows:
In this paper, the authors propose a novel framework to combine the strength of both LLMs and domain models to better utilize the precedents for the legal judgment prediction (LJP) task. Specifically, domain models contribute by providing candidate labels and finding the proper precedents from the case database effectively; the LLMs will decide the final prediction through an in-context precedent comprehension. Experiments on the real-world dataset demonstrate the effectiveness of the proposed model.

**Questions For The Authors:**

 This paper can be improved according to the following aspects.
1. The authors should compare with the relevant work in 2022 to prove the effectiveness of the proposed method, such as TOPJUDGE (Feng et al., 2022, Legal Judgment Prediction via Event Extraction with Constraints) and CTM (Liu et al., 2022, Augmenting Legal Judgment Prediction with Contrastive Case Relations).
2. Following LADAN (Xu et al., 2020) and R-Former (Dong and Niu, 2021), the authors should also verify the model using the same datasets as the previous method, such as CAIL-small and CAIL-big.
3. The proposed method is a pipeline model, and the final prediction results of the LLMs are heavily dependent on the candidate results of the first-stage prediction model. Could the authors further analyze the reliability of the candidate results?
4. In the ablation experiment, when the LLMs is forced to generate the explanation of the prediction, the performance of the model is greatly affected, with both improvement and decrease. Could the authors conduct further detailed analysis?


**Reasons To Accept:**

1. The authors effectively utilize the precedents information to enhance the performance of the model.
2. The proposed method combines the strength of both LLMs and domain models, and provides a promising direction for the collaboration of LLMs and domain-model.



**Reasons To Reject:**

1. The authors should compare with the relevant work in 2022 to prove the effectiveness of the proposed method, such as TOPJUDGE (Feng et al., 2022, Legal Judgment Prediction via Event Extraction with Constraints) and CTM (Liu et al., 2022, Augmenting Legal Judgment Prediction with Contrastive Case Relations).
2. Following LADAN (Xu et al., 2020) and R-Former (Dong and Niu, 2021), the authors should also verify the model using the same datasets as the previous method, such as CAIL-small and CAIL-big.
3. The proposed method is a pipeline model, and the final prediction results of the LLMs are heavily dependent on the candidate results of the first-stage prediction model. Could the authors further analyze the reliability of the candidate results?
4. In the ablation experiment, when the LLMs is forced to generate the explanation of the prediction, the performance of the model is greatly affected, with both improvement and decrease. Could the authors conduct further detailed analysis?


**Reproducibility:**

3: Could reproduce the results with some difficulty. The settings of parameters are underspecified or subjectively determined; the training/evaluation data are not widely available.

**Reviewer Confidence:**

4: Quite sure. I tried to check the important points carefully. It's unlikely, though conceivable, that I missed something that should affect my ratings.

---

> ### Author Rebuttal · Authors · 2023-08-29
>
> Thank you for dedicating your time to reviewing our paper and offering us valuable insightful feedback. We are grateful for your recognition of the novelty and potential of our idea. With regard to the concerns you raised, we have carefully considered them and would like to address them in the following response.
>
> Q1:
>
>     For the concern about "update baselines":
>
> - We have implemented a comparison of these two baseline models EPM[1] and CTM[2], and you can find the results displayed in the following tables.
>
> **Task: Law article prediction**
>
> #### Dataset: CJO22
>
> | Method     | Acc       | Ma-P      | Ma-R      | Ma-F      |
> | ---------- | --------- | --------- | --------- | --------- |
> | EPM        | 84.19     | 47.21     | 43.79     | 44.39     |
> | CTM        | 79.44     | 47.83     | 42.25     | 43.43     |
> | PLJP(CNN)  | 87.67     | 55.21     | 55.59     | 54.37     |
> | PLJP(BERT) | **94.18** | **74.65** | **76.23** | **78.84** |
>
> #### Dataset: CAIL-2018
>
> | Method     | Acc       | Ma-P      | Ma-R      | Ma-F      |
> | ---------- | --------- | --------- | --------- | --------- |
> | EPM        | 85.80     | 49.08     | 45.76     | 47.32     |
> | CTM        | 84.72     | 46.46     | 44.83     | 45.10     |
> | PLJP(CNN)  | 86.05     | 58.08     | 56.46     | 54.92     |
> | PLJP(BERT) | **87.07** | **58.81** | **57.29** | **56.63** |
>
> **Task: Charge prediction**
>
> #### Dataset: CJO22
>
> | Method     | Acc       | Ma-P      | Ma-R      | Ma-F      |
> | ---------- | --------- | --------- | --------- | --------- |
> | EPM        | 83.49     | 80.36     | 83.29     | 81.87     |
> | CTM        | 79.33     | 82.39     | 83.12     | 82.81     |
> | PLJP(CNN)  | 91.62     | 83.43     | 84.88     | 83.40     |
> | PLJP(BERT) | **94.18** | **90.25** | **88.67** | **89.05** |
>
> #### Dataset: CAIL-2018
>
> | Method     | Acc       | Ma-P      | Ma-R      | Ma-F      |
> | ---------- | --------- | --------- | --------- | --------- |
> | EPM        | 91.20     | 90.81     | 89.99     | 90.46     |
> | CTM        | 90.28     | 90.34     | 88.08     | 86.30     |
> | PLJP(CNN)  | 91.49     | 81.80     | 83.95     | 80.06     |
> | PLJP(BERT) | **94.99** | **92.12** | **91.10** | **91.33** |
>
> **Task: Prison term prediction**
>
> #### Dataset: CJO22
>
> | Method     | Acc       | Ma-P      | Ma-R      | Ma-F      |
> | ---------- | --------- | --------- | --------- | --------- |
> | EPM        | 36.91     | 30.65     | 31.61     | 30.20     |
> | CTM        | 36.81     | 27.10     | 25.96     | 26.46     |
> | PLJP(CNN)  | 36.51     | 20.21     | 21.44     | 20.07     |
> | PLJP(BERT) | **43.52** | **33.37** | **35.67** | **31.98** |
>
> #### Dataset: CAIL-2018
>
> | Method     | Acc       | Ma-P      | Ma-R      | Ma-F      |
> | ---------- | --------- | --------- | --------- | --------- |
> | EPM        | 40.25     | 37.96     | 37.00     | 37.34     |
> | CTM        | 39.56     | 38.66     | **38.02** | **37.84** |
> | PLJP(CNN)  | 40.81     | 32.77     | 35.59     | 25.71     |
> | PLJP(BERT) | **48.72** | **42.64** | 36.80     | 35.43     |
>
> - The results prove the effectiveness of our PLJP method, especially in the CJO22 dataset. We will add this result and host additional analysis in the revision to help readers better consume the experiment outcomes.
>
> Q2:
>
>     For the concern about "datasets":
>
> - Thanks for the suggestion. In our experiments, we utilized two datasets: CAIL-2018 and CJO22. The CAIL-2018 used in our paper is the same one referenced as CAIL-small. Additionally, the CJO22 dataset was constructed to specifically contain cases that took place post 2022. This was done to prevent data leakage from occurring during training since all LLMs were trained using a corpus collected prior to September 2021.
>
> - In addition, we are conducting experiments on the CAIL-big dataset, which is derived from the same resource as the CAIL-small. Unfortunately, due to the substantial size of the CAIL-big dataset and the limited inference speed of ChatGPT, obtaining immediate experimental results during the rebuttal period is challenging. We intend to include the results for the CAIL-big dataset in the revised version of our work.
>
> - Importantly, we will release all the data and code to ensure the reproducibility of our work.
>
> Q3:
>
>     For the concern about "reliability of the candidate results":
>
> - We report the top-5 accuracy of the two domain models (e.g., CNN and BERT) here:
>
> **Dataset: CAIL-2018**
>
> #### Method: CNN
>
> #### Task: Law article prediction
>
> | topk | Acc   | Ma-P  | Ma-R  | Ma-F  |
> | ---- | ----- | ----- | ----- | ----- |
> | 1    | 80.50 | 40.10 | 38.33 | 38.49 |
> | 2    | 91.20  | 75.67 | 76.14 | 75.34 |
> | 3    | 93.52 | 78.08 | 79.04 | 78.30  |
> | 4    | 94.88 | 81.14 | 81.97 | 81.31 |
> | 5    | 95.28 | 81.99 | 82.25 | 81.92 |
>
> #### Task: Charge prediction
>
> | topk | Acc   | Ma-P  | Ma-R  | Ma-F  |
> | ---- | ----- | ----- | ----- | ----- |
> | 1    | 87.52 | 88.23 | 88.31 | 88.17 |
> | 2    | 94.48 | 94.62 | 94.44 | 94.46 |
> | 3    | 95.84 | 95.97 | 95.79 | 95.82 |
> | 4    | 97.12 | 97.22 | 97.10  | 97.11 |
> | 5    | 97.92 | 97.97 | 97.94 | 97.93 |
>
> #### Task: Prison term prediction
>
> | topk | Acc   | Ma-P  | Ma-R  | Ma-F  |
> | ---- | ----- | ----- | ----- | ----- |
> | 1    | 34.42 | 32.22 | 30.53 | 31.05 |
> | 2    | 53.04 | 51.22 | 49.55 | 49.73 |
> | 3    | 66.56 | 64.91 | 63.59 | 63.53 |
> | 4    | 78.56 | 76.94 | 75.94 | 75.86 |
> | 5    | 88.16 | 87.39 | 86.64 | 86.71 |
>
> #### Method: BERT
>
> #### Task: Law article prediction
>
> | topk | Acc   | Ma-P  | Ma-R  | Ma-F  |
> | ---- | ----- | ----- | ----- | ----- |
> | 1    | 82.77 | 36.82 | 35.94 | 35.82 |
> | 2    | 93.04 | 77.21 | 76.75 | 76.55 |
> | 3    | 95.20  | 83.27 | 82.8  | 82.67 |
> | 4    | 96.00  | 85.95 | 84.71 | 85.06 |
> | 5    | 96.64 | 89.07 | 87.8  | 88.16 |
>
> #### Task: Charge prediction
>
> | topk | Acc   | Ma-P  | Ma-R  | Ma-F  |
> | ---- | ----- | ----- | ----- | ----- |
> | 1    | 89.10 | 90.10 | 89.48 | 89.63 |
> | 2    | 95.12 | 95.39 | 95.00  | 95.09 |
> | 3    | 96.32 | 96.49 | 96.23 | 96.30  |
> | 4    | 97.36 | 97.51 | 97.26 | 97.33 |
> | 5    | 97.76 | 97.87 | 97.7  | 97.75 |
>
> #### Task: Prison term prediction
>
> | topk | Acc   | Ma-P  | Ma-R  | Ma-F  |
> | ---- | ----- | ----- | ----- | ----- |
> | 1    | 40.00 | 37.53 | 33.66 | 33.58 |
> | 2    | 65.20  | 66.03 | 57.89 | 59.65 |
> | 3    | 80.32 | 81.37 | 73.90  | 76.60  |
> | 4    | 89.28 | 90.26 | 84.20  | 86.58 |
> | 5    | 94.96 | 94.92 | 90.67 | 92.48 |
>
> **Dataset: CJO22**
>
> #### Method: CNN
>
> #### Task: Law article prediction
>
> | topk | Acc   | Ma-P  | Ma-R  | Ma-F  |
> | ---- | ----- | ----- | ----- | ----- |
> | 1    | 76.14 | 35.48 | 38.55 | 35.39 |
> | 2    | 84.39 | 42.99 | 47.24 | 43.18 |
> | 3    | 87.51 | 46.68 | 50.92 | 47.40  |
> | 4    | 89.22 | 50.38 | 55.32 | 51.50  |
> | 5    | 90.28 | 52.25 | 57.42 | 53.54 |
>
> #### Task: Charge prediction
>
> | topk | Acc   | Ma-P  | Ma-R  | Ma-F  |
> | ---- | ----- | ----- | ----- | ----- |
> | 1    | 74.91 | 74.00 | 78.12 | 73.97 |
> | 2    | 85.10  | 84.16 | 87.29 | 84.20  |
> | 3    | 88.16 | 87.73 | 90.5  | 87.91 |
> | 4    | 90.28 | 90.08 | 92.49 | 90.24 |
> | 5    | 91.87 | 91.95 | 94.26 | 92.31 |
>
> #### Task: Prison term prediction
>
> | topk | Acc   | Ma-P  | Ma-R  | Ma-F  |
> | ---- | ----- | ----- | ----- | ----- |
> | 1    | 27.38 | 18.48 | 17.51 | 17.44 |
> | 2    | 44.88 | 35.42 | 32.78 | 33.19 |
> | 3    | 61.25 | 51.03 | 48.92 | 49.08 |
> | 4    | 71.97 | 62.22 | 60.24 | 59.72 |
> | 5    | 81.39 | 70.73 | 68.63 | 68.54 |
>
> #### Method: BERT
>
> #### Task: Law article prediction
>
> | topk | Acc   | Ma-P  | Ma-R  | Ma-F  |
> | ---- | ----- | ----- | ----- | ----- |
> | 1    | 82.62 | 45.89 | 47.91 | 45.83 |
> | 2    | 90.16 | 53.31 | 55.48 | 53.42 |
> | 3    | 93.17 | 61.03 | 63.42 | 61.65 |
> | 4    | 94.58 | 65.16 | 66.50  | 65.27 |
> | 5    | 95.23 | 66.98 | 68.59 | 67.26 |
>
> #### Task: Charge prediction
>
> | topk | Acc   | Ma-P  | Ma-R  | Ma-F  |
> | ---- | ----- | ----- | ----- | ----- |
> | 1    | 80.50 | 80.34 | 81.09 | 78.36 |
> | 2    | 89.58 | 89.84 | 90.36 | 88.52 |
> | 3    | 92.87 | 93.45 | 94.06 | 92.97 |
> | 4    | 94.17 | 94.72 | 95.50  | 94.61 |
> | 5    | 95.11 | 95.7  | 96.21 | 95.57 |
>
> #### Task: Prison term prediction
>
> | topk | Acc   | Ma-P  | Ma-R  | Ma-F  |
> | ---- | ----- | ----- | ----- | ----- |
> | 1    | 36.80 | 29.83 | 27.50 | 27.03 |
> | 2    | 59.42 | 53.47 | 47.95 | 48.53 |
> | 3    | 73.79 | 65.66 | 59.86 | 61.30 |
> | 4    | 85.45 | 76.11 | 70.79 | 72.57 |
> | 5    | 92.40 | 81.95 | 78.72 | 79.94 |
>
> - Examining the results provided, it becomes evident that the improvement in accuracy from the top1 to top3 predictions is more substantial compared to the improvement from top3 to top5 predictions. Take the law article prediction of BERT in CJO22 as an example, the top1 accuracy stands at 82.62%, the accuracy for the top3 predictions increases to 93.17%, while the top5 accuracy is 95.23%.
>
> - In our PLJP framework, every candidate label necessitates a corresponding precedent. Due to the constraint of prompt length, we made the decision to select the top3 labels as the candidate labels for our experiments. We will add a detailed analysis of the above results in the revision.
>
> Q4:
>
>     For the concern about "model performance with explanation":
>
> - Through our ablation experiments, we find that the prediction accuracy of PLJP is affected when we force the LLM to generate the explanation of the prediction simultaneously. This may be due to a "trade-off" between accuracy and interpretability.
> - A straightforward approach to maintain prediction accuracy is to have the LLM generate the explanation in a separate dialogue round following the prediction, rather than do the prediction and the explanation simultaneously.
> - In the future, we will explore this interesting problem and try to enhance the predictions with explanation.
>
>
>
> Missing Reference:
>
> Thanks for the reminder, we will add the missing references [1-2].
>
> [1]Dugang Liu, Weihao Du, Lei Li, Weike Pan, and Zhong Ming. 2022. Augmenting legal judgment prediction with contrastive case relations. In Proceedings of the 29th International Conference on Computational Linguistics, pages 2658–2667, Gyeongju, Republic of Korea. International Committee on Computational Linguistics.
>
> [2]Yi Feng, Chuanyi Li, and Vincent Ng. 2022. Legal judgment prediction via event extraction with constraints. In Proceedings of the 60th Annual Meeting of the Association for Computational Linguistics (Volume 1: Long Papers), pages 648–664, Dublin, Ireland. Association for Computational Linguistics.

---

### Meta-Review · Area_Chair_qudR · 2023-09-12

**Recommendation:** 4

**Metareview:**

The paper proposes an interesting retrieval-augmented approach utilizing LLMs for the task of legal judgement prediction.
The proposed method combines the strength of both LLMs and domain models, and provides a promising direction for the collaboration of LLMs and domain-model.

The specific doubts of reviewer ZR8e have been addressed in the author response.

---

### Decision · Program_Chairs · 2023-10-07

**Decision:**

Accept-Main

**Comment:**

The paper proposes an interesting retrieval-augmented approach utilizing LLMs for the task of legal judgement prediction.
The proposed method combines the strength of both LLMs and domain models, and provides a promising direction for the collaboration of LLMs and domain-model.

The specific doubts of reviewer ZR8e have been addressed in the author response.